# The Future of HER2-Targeted Treatment for Osteosarcoma: Lessons from the Negative Trastuzumab Deruxtecan Results

**DOI:** 10.3390/ijms242316823

**Published:** 2023-11-27

**Authors:** Kenji Nakano

**Affiliations:** Department of Medical Oncology, Cancer Institute Hospital of Japanese Foundation for Cancer Research, 3-8-31 Ariake, Koto, Tokyo 135-8550, Japan; kenji.nakano@jfcr.or.jp; Tel.: +81-3-3520-0111

**Keywords:** osteosarcoma, targeted therapy, HER2, trastuzumab deruxtecan, antibody–drug conjugate, CAR-T

## Abstract

Human epidermal growth factor receptor 2 (HER2), coded by the proto-oncogene *ERBB*, is known to be mutated or amplified in various malignant diseases, and many HER2-targeted therapies (including monoclonal antibodies and low-molecular-weight tyrosine kinase inhibitors) have been investigated. HER2 overexpression is observed in ~30% of patients with osteosarcoma, and HER2-targeted therapy for osteosarcoma has also been investigated, along with the prognostic and/or predictive value of HER2. An effective HER2-targeted therapy for osteosarcoma has not been established, however. An antibody–drug conjugate (ADC), i.e., trastuzumab deruxtecan (T-DXd), has been approved for the treatment of HER2-positive malignant diseases such as breast cancer and gastric cancer. T-DXd showed promising efficacy in a tumor-agnostic clinical trial, but even T-DXd did not demonstrate sufficient efficacy against HER2-positive osteosarcoma. In this review, the underlying reasons/mechanisms for the failure of HER2-targeted treatments for osteosarcoma (including T-DXd) are discussed, and the potential and future direction of HER2-targeted therapy is described.

## 1. Development of HER2-Targeted Therapy in Oncology

### 1.1. HER2 Biology

Human epidermal growth factor receptor 2 (HER2), coded by the proto-oncogene *ERBB2* (located at chromosome 17q21), is a receptor tyrosine-protein kinase and a member of the family of epidermal growth factor receptor (EGFR) tyrosine kinases, which consists of EGFR (ErbB1), HER2 (ErbB2), HER3 (ErbB3), and HER4 (Erb4). Unlike other EGFR family members, specific ligand binding to HER2 is not known; HER2 acts to form homodimers when binding to other HER2 molecules or heterodimers when binding to other EGFR family members, activating downstream cell proliferation signaling pathways such as the mitogen-activated protein kinase (MAPK) and phosphatidylinositol bisphosphate 3 kinase (PI3K) pathways [1]. The role of overexpressed *ERBB2* as an oncogene was clarified with the use of transgenic mouse models [2].

The most common cancer known to include an HER2-positive population is breast cancer, for which HER2-targeted treatment strategies have been established [3]. As with the treatment strategies, the pathological evaluation of HER2 overexpression has been standardized for clinical decisions regarding the indications for HER2-targeted therapy [4], and pathological evaluation of HER2 overexpression is now applied to other malignant diseases. Gastroesophageal adenocarcinoma includes an HER2-positive subtype, and its treatment strategies have been divided into those for HER2-positive versus HER2-negative populations [5]. Although less common than breast and gastroesophageal cancers, a number of patients with other carcinomas have been observed to be HER2-positive (Table 1) [1,6,7]. As next-generation sequence (NGS) and whole-genome profiling have become widely used in clinical practice, various rare HER2 mutations other than amplification/overexpression have been recognized, such as *ERBB2* fusion genes [8].

### 1.2. Development of HER2-Targeted Drugs

Based on the results obtained in basic research and the findings described above, many HER2-targeted drugs have been investigated and approved over the past 20 years, including monoclonal antibodies (mAbs), low-molecular-weight tyrosine kinase inhibitors (TKIs), and antibody–drug conjugates (ADCs). Most of these drugs were approved first for the treatment of breast cancer, and some of them have shown clinical benefits for other malignant diseases and are approved or recommended in some guidelines. The HER2-targeted drugs that have been approved by the U.S. Food and Drug Administration (FDA) as of 2023 are listed in Table 2.

Trastuzumab is the pioneer of molecular-targeted drugs; it was approved by the FDA in 1998 and was the first mAb used to treat malignant diseases. Trastuzumab has been shown to be useful in a wide range of circumstances, including recurrent cases and pre- and postoperative adjuvant chemotherapy for HER2-positive breast cancer, making it an indispensable drug in clinical practice [3]. Trastuzumab has also been approved by the FDA for the treatment of HER2-positive gastroesophageal cancer in combination with chemotherapy for recurrent or metastatic cases [9]. Several clinical trials have evaluated trastuzumab’s efficacy against other cancers, and guideline recommendations or indication approval by the FDA have been applied for some cancers, including colorectal cancer and salivary gland cancer [6,10]. Trastuzumab is a drug of great significance in cancer treatment in the sense that it opened up the therapeutic system of HER2-targeted therapy. Along with the widespread use of trastuzumab and the increased demand for cancer treatment, a trastuzumab biosimilar was developed and received FDA approval in 2017 [11,12]. A convenient subcutaneous injection formulation of trastuzumab is also being developed [13,14].

Pertuzumab is a mAb that targets the binding site of dimers including HER2, and it is usually administered in combination with trastuzumab. Pertuzumab has been shown to be effective in rescue therapy and adjuvant chemotherapy for recurrent and metastatic breast cancer [15,16]. The efficacy of pertuzumab in combination with trastuzumab against malignant diseases other than breast cancer is under investigation [17], and subcutaneous formulations have been developed and approved, as with trastuzumab [18]. The mAb margetuximab has shown benefit with cytotoxic chemotherapy for HER2-positive breast cancer patients with a history of multiple chemotherapies including other HER2-targeted therapy [19,20].

Lapatinib, neratinib, and tucatinib are oral low molecular TKIs that target HER2 and other EGFR-family receptor tyrosine kinases. Lapatinib has demonstrated antitumor activity in combination with the cytotoxic anticancer drug capecitabine and is approved for the treatment of breast cancer [21]. Neratinib is also used with capecitabine to treat recurrent/metastatic breast cancer, but it is used as monotherapy in extended adjuvant therapy; for perioperative therapy with neratinib, the selection of indications as precision medicine based on patient profiles has been investigated [22,23,24]. Tucatinib is prescribed in combination with capecitabine and trastuzumab, and it has demonstrated a promising effect on brain metastases [25,26]. Tucatinib and trastuzumab combination therapy was also approved for HER2-positive colorectal cancer in 2023 [27].

Trastuzumab emtansine (T-DM1) is an antibody–drug conjugate (ADC) in which the microtubule inhibitor DM1 is bound as a cytotoxic payload by the linker to trastuzumab. T-DM1 showed efficacy for patients with advanced HER2-positive breast cancer after trastuzumab treatment, and it was superior to lapatinib and capecitabine [28]. As perioperative therapy, T-DM1 is administered to patients who have residual invasive disease after neoadjuvant chemotherapy including trastuzumab [29].

All of the HER2-targeted drugs mentioned above are now approved for the treatment of breast cancer, but their efficacy against other HER2-positive cancers varies; although there has been a study in which HER2-targeted therapy showed promising efficacy similar to that of trastuzumab for salivary gland cancers [10], there is also the case that HER2-targeted therapy effective to HER2-positive breast cancer is not effective to other HER2-positive cancers, such as T-DM1 to gastroesophageal cancer [30]. Until the introduction of trastuzumab deruxtecan (T-DXd, discussed below in Section 3), the prognostic impact of HER2 expression and the efficacy of targeted therapy was essentially considered tumor specific, not tumor agnostic [6].

## 2. HER2 in Osteosarcoma

### 2.1. HER2 Amplification or Overexpression in Osteosarcoma

Osteosarcoma is a malignant disease that occurs mainly in children and young adults. It is the most common primary bone tumor but it is rare; for example, there are approx. 400 new osteosarcoma cases/year in the United States. Since the 1980s, the use of cytotoxic anticancer agents (methotrexate, doxorubicin, cisplatin, and ifosfamide) for pre- and postoperative adjuvant chemotherapy has improved the prognoses of osteosarcoma patients, but the development of molecular-targeted therapy for osteosarcomas has lagged behind that of other solid tumors [31]. One reason for this is that the genetic variation in osteosarcomas is so great that it is difficult to find a common specific target molecule or mutation [32]. Among the osteosarcomas, mutations to HER2 have been detected at a relatively high rate, but the reported frequency of HER2 amplification or overexpression in osteosarcoma varies (although most reports are in the 30–60% range, there have been reports of 0–100%). This wide range of values may depend on the evaluation method (e.g., immunohistochemistry [IHC] techniques such as staining vs. fluorescence in situ hybridization [FISH]) and patient background (initial presentation, distant metastasis, etc.) [33]. It has been noted that HER2 tends to be expressed in the cytoplasm in osteosarcoma cell lines [34]. This differs from HER2-positive breast cancer and other solid tumors, which may also contribute to the positivity rate’s variability.

Since HER2 is known to have carcinogenic function, the value of HER2 mutation as a prognostic factor in osteosarcoma has been investigated, but the prognostic significance of HER2 in osteosarcoma is still a matter of controversy [33].

### 2.2. HER2-Targeted Therapies for Osteosarcoma

There are very few clinical data on HER2-targeted therapies for osteosarcoma compared to other carcinomas, due in part to its rarity; in addition, osteosarcoma is predominant in children and young adults, and safety and optimal dosage evaluations for HER2-targeted drugs must be considered separately in the development of such drugs.

As mentioned above, trastuzumab is the first and representative HER2-targeted antitumor agent, and it is the only HER2-targeted agent whose efficacy has been evaluated in prospective clinical trials for osteosarcoma prior to the advent of trastuzumab deruxtecan. In the phase II trial carried out by the Children’s Oncology Group, 96 newly diagnosed metastatic osteosarcoma patients (41 with HER2 overexpression shown by IHC) received both standard chemotherapy (a combination of cisplatin, doxorubicin, methotrexate, ifosfamide, and etoposide); trastuzumab (weekly for a total of 34 cycles) was added to the treatment of 34 of the HER2-positive patients. The 30-month event-free survival and overall survival rates of the HER2-positive patients were 32% and 59%, respectively, which were not significantly different from the rates of the HER2-negative patients (32% and 50%) [35]. Although the efficacy of trastuzumab cannot be evaluated in a randomized controlled trial since the study did not directly compare the efficacy of trastuzumab with or without trastuzumab in HER2-positive patients, the results of the Children’s Oncology Group study did not indicate that trastuzumab would be effective for the treatment of HER2-positive osteosarcoma. Other than the phase II trial of trastuzumab cited above, data from prospective clinical trials of HER2-targeted therapies have not been available, and preclinical studies remain few and far between [36].

It was against this backdrop that a new HER2-targeted ADC, i.e., trastuzumab deruxtecan (T-DXd) emerged, as discussed next.

## 3. Trastuzumab Deruxtecan (T-DXd) for the Treatment of Osteosarcoma

### 3.1. The Mechanism of Action and Clinical Efficacy of T-DXd

Trastuzumab deruxtecan (T-DXd) is a new HER2-targeted ADC with a novel linker-payload system. The cytotoxic payload is a topoisomerase I inhibitor exatecan derivative (DXd), which differs from the cytotoxic payload of a previous HER2-targeted ADC, i.e., T-DM1, in which the microtubule inhibitor is linked as payload [37]. Due to the cell membrane permeability effect, there is not only a direct cytotoxic effect of T-DXd against the target HER2-positive cancer cells to which trastuzumab directly binds as an HER2-targeted antibody; T-DXd also has the ability to kill cancer cells in the vicinity, as a so-called bystander effect [38]. A preclinical study of T-DXd demonstrated the bystander effect, which is considered to be a characteristic of this drug [39]. As expected from the results of preclinical studies, T-DXd has shown promising effects on various HER2-positive solid tumors, which had not been regarded as the subjects of HER2-targeted therapies in the early phase of investigation [40].

First, in HER2-positive breast cancer, T-DXd treatment produced a very high response rate of 60% in patients who had received multiple HER2-targeted therapies, including trastuzumab and T-DM1, and T-DXd was then approved by the FDA for the treatment of HER2-positive breast cancer in 2019 [41]. In a subsequent phase III trial, T-DXd treatment provided extended survival in patients after T-DM1 treatment [X], and in another phase III trial T-DXd showed superiority to T-DM1 [42]. Based on these results, T-DXd is gradually being used in previous lines of therapy in patients with recurrent or metastatic disease. T-DXd has also shown efficacy against trastuzumab-resistant HER2-positive gastroesophageal cancer, and it was approved by the FDA for the treatment of this disease in 2021 [43]. In addition, high response rates have been reported for T-DXd in settings in which HER2-targeted therapy had not been approved. Clinical trials for HER2-positive patients with non-small cell lung cancer (NSCLC) and HER2-positive patients with colorectal cancer have shown high response rates [44,45], and T-DXd was approved by the FDA for the treatment of NSCLC in 2022. Other clinical trials are underway for gynecologic cancer [46] and biliary tract cancer [47]. In a single-arm meta-analysis of T-DXd as a treatment for HER2-positive solid tumors (breast cancer, gastroesophageal cancer, colorectal cancer, NSCLC, and biliary tract cancer), the pooled overall response rate was 47.91%, and the median durations of progression-free and overall survival combined were 9.63 months and 10.71 months, respectively [48].

As noted above, the efficacy of HER2-targeted therapy has been thought to vary between tumors of different primary origin, but the results demonstrating efficacy in such a wide range of areas has raised expectations for the tumor-agnostic effects of T-DXd [49]. In 2023, the DESTINY-PanTumor02 study was initiated as a phase II trial of T-DXd enrolling HER2-positive (IHC 2+ or IHC 3+) solid tumors for which HER2-targeted therapies had rarely or never been evaluated, including biliary tract, bladder, cervical, endometrial, ovarian, pancreatic, and other tumors [50]. The trial’s overall response rate (primary endpoint) for all patients (IHC 3+ and IHC 2+) and IHC 3+ was 37.1% and 61.3%, respectively. There were variations in the efficacy of T-DXd among the types of carcinoma, with pancreatic cancer, in particular, having an exceptionally low response rate at 4.0%, but, in general, a high response rate was achieved with T-DXd in each of the HER2-positive tumors enrolled, regardless of their primary origins.

Until recently, HER2-targeted therapy had been known to be more effective in cases with higher HER2 expression, and its efficacy was regarded as clinically meaningful only for patients with high HER2 expression (IHC 3+ or FISH-positive, defined as “HER2-positive”), and thus the clinical trials of HER2-targeted therapies before the advent of T-DXd had been limited to these patients. This was also true for T-DM1, the first HER2-targeted ADC. However, T-DXd has been shown to be effective even in patients with low HER2 expression, and its efficacy was demonstrated in clinical trials of breast cancer patients with low HER2 expression; its approval has established a new population called “HER2-low” [51]. Clinical trials have also been conducted in patients with other cancers with low HER2 expression, and promising results have been obtained [52].

Although a high response rate and clinical benefit have been observed for T-DXd in a broad range of malignant diseases, the management of adverse events is a major challenge, especially in interstitial lung disease (ILD) [53,54].

### 3.2. T-DXd Treatment for Osteosarcoma—An Unexpected Result

The above-described background has provided an opportunity to evaluate the efficacy of T-DXd in osteosarcoma, which is known to contain HER2-positive cases. However, HER2-targeted therapies have not been developed for osteosarcoma. In a preclinical study conducted by the Pediatric Preclinical Testing Consortium, six of seven xenograft models of osteosarcoma with HER2 mRNA positivity achieved prolonged event-free survival, suggesting that clinical trials may be beneficial [55].

Nevertheless, the results of the PEPN1924 phase II trial of T-DXd were disappointing. Recurrent, unresectable osteosarcoma patients with >10% of tumor cells with cytoplasmic or membranous HER2 expression were treated with T-DXd (5.4 mg/kg every 3 weeks), but none of nine patients enrolled at the first step showed an objective response; on the contrary, seven of them were evaluated as having progressive disease at the first response evaluation (after only two infusions of T-DXd), and one of the remaining two patients withdrew consent before the first evaluation; the other barely reached the definition of stable diseases at 24 weeks after the introduction of T-DXd, which was the primary endpoint of the trial. As a result, the trial was terminated without proceeding to the second step [56].

Why was T-DXd, which has been shown to be effective in almost all HER2-positive cancers, not effective in osteosarcoma? Several reasons should be considered. An initial question is, were the inclusion criteria of the PEPN1924 trial appropriate? The definition of the inclusion criteria for “HER2-positive” status in the PEPN1924 trial differs from the definition used for other HER2-positive cancers (represented by breast cancer). As discussed earlier in Section 2.1, HER2 in osteosarcoma tends to be expressed in the cytoplasm rather than on the cell membrane surface, which may have influenced the efficacy of T-DXd. In preclinical studies, T-DXd was shown to prolong event-free survival in the xenograft model, but the effect of tumor shrinkage was limited to stable disease [55]. However, it is expected that T-DXd will have an antitumor bystander effect, and we can judge the degree to which the HER2 expression status affects the efficacy of T-DXd only by considering the results of past and future clinical trials.

Is it possible that the antitumor effect of the payload in T-DXd is not active against osteosarcoma? The payload of T-DXd is an exatecan derivative (DC-8951 derivative, DXd), which is a topoisomerase I inhibitor similar to irinotecan [37], but irinotecan is not approved for the treatment of osteosarcoma. In a retrospective analysis of patients treated at the Memorial Sloan-Kettering Cancer Center in which the efficacy of irinotecan for treating pediatric solid tumors was evaluated, seven osteosarcoma patients (six of whom had an evaluable response) were included and none of them showed any clinical benefit; the outcome for all six evaluable patients was progressive disease [57]. However, the efficacy expected from the mechanism of action of the payload does not necessarily match the efficacy of the ADC. For example, topoisomerase inhibitors including irinotecan have not been approved for breast cancer (for which T-DXd is markedly effective), and conversely, there are carcinomas such as pancreatic cancer for which irinotecan is used as standard therapy (including liposomal formulations), but T-DXd was not effective [58].

Whether the dose of T-DXd is appropriate is also open to consideration. Several doses of T-DXd are currently approved for use: 6.4 mg/kg and 5.4 mg/kg, depending on the carcinoma. The PEPN1924 trial used the 5.4 mg/kg dose, but it is unclear whether the dose intended for adults is effective and safe in children, which is especially important because osteosarcomas have developed in many pediatric patients. The majority of the patients in the PEPN1924 trial ended the study with early progression, and it is thus more likely that underdosing resulted in a lack of efficacy rather than in adverse events due to overdosing. However, T-DXd is a drug with a high incidence of adverse events—especially pulmonary toxicity as typified by ILD—and a carefully designed phase I trial is necessary if efficacy with increased doses is to be achieved.

Another possibility to consider is that the role that HER2 plays in the cancer cell signaling pathway in osteosarcoma may play a different role than in other HER2-positive solid tumors. For example, BRAF V600 mutation is now approved as the tumor-agnostic target [59,60], but differences in efficacy are observed among tumor origins, especially in colorectal cancer. Interestingly, unlike other BRAF-mutated cancers, it is known that the co-inhibition of EGFR (upstream of BRAF) is indispensable to the clinical response of colorectal cancers, and, on the other hand, the co-inhibition of MEK (downstream of BRAF) might not contribute to the improved efficacy in BRAF-mutated colorectal cancers [61,62]. Indeed, regorafenib, a multi-kinase inhibitor, has shown some efficacy in osteosarcoma, suggesting that multiple cell surface tyrosine kinases and their downstream signaling pathways are involved in tumor growth [63,64]. Investigations of how other signaling pathways associated with HER2 are involved in cancer progression may help identify osteosarcoma patients who will benefit from T-DXd in the future.

In any case, the evidence available at present does not indicate that T-DXd is effective against HER2-positive osteosarcoma, and it is not a drug that will change the treatment system as significantly as other malignant diseases.

## 4. Future Direction of HER2-Targeted Therapy Development to Osteosarcoma

Although HER2-targeted therapies (i.e., mAbs and ADCs) have made great strides in cancer treatment with results extending beyond breast cancer, osteosarcoma has not yet benefited from them. New therapeutic strategies targeting HER2 are still being developed, some of which target osteosarcoma. The next section is an overview of the prospects for HER2-targeted therapy as a potential new treatment for osteosarcoma (Figure 1).

### 4.1. Bispecific Antibody Including HER2

The development of bispecific antibodies that inhibit multiple targets is progressing, with practical approval achieved in hematologic diseases [65,66,67]. Although bispecific antibodies for solid tumors are still under development and appropriate target molecules have not been established, HER2, for which molecular-targeted therapy with antibody drugs has already been established as discussed above, is one of the candidates [68,69].

Another potential candidate for treating osteosarcoma is disialoganglioside (GD2), which is a glycosphingolipid expressed on the surface of various malignant tumors including osteosarcoma [70]. The development of GD2-targeted therapy has focused on pediatric oncology, including neuroblastomas and osteosarcomas; as a treatment for neuroblastoma, the anti-GD2 antibody dinutuximab showed the improvement of event-free survival as maintenance therapy with alternating granulocyte-macrophage colony-stimulating factor (GM-CSF) and intravenous interleukin-2 (IL-2) and was approved by the FDA in 2015 [71]. Unfortunately, dinutuximab for recurrent osteosarcoma did not show efficacy in a phase II trial [72], but the development of bispecific antibodies that inhibit HER2 and GD2 is underway, and their efficacy is expected to be evaluated [73]. Attempts have also been made to evaluate the GD2 expression on images [74], which is a technique that is beneficial in osteosarcoma, a primary bone cancer that can be more difficult to evaluate with the use of molecular pathologic markers compared to other solid tumors. As described in the example of BRAF-targeted therapy for colorectal cancer, it is possible that a combination of targeted agents can inhibit cancer growth even if individual targeted therapies do not work. However, in the development of bispecific antibodies, clinical research should not be conducted by combining targets in the dark; basic and translational research should also be conducted to determine how much of a synergy effect can be expected from co-targeted therapy.

### 4.2. Immunotherapy Targeted to HER2 in Osteosarcoma

Cancer immunotherapy progressed remarkably in the 2010s, due mainly to the development of immune checkpoint inhibitors [75]. However, for bone and soft tissue sarcomas, the immune checkpoint inhibitors have not yet shown meaningful clinical benefit, except for some specific subtypes of soft tissue sarcomas [76,77]. Immunotherapies such as T-cell treatments and vaccines for bone and soft tissue tumors continue to be developed, some of which target HER2.

Chimeric antigen receptor T-cell (CAR-T) therapy treats cancer by taking the patient’s own T cells and administering them to the patient with modified T cells (CAR-T cells) that can produce a special protein called chimeric antigen receptor (CAR) [78]. Formulations targeting the transmembrane protein CD19 and B-cell maturation antigen (BCMA) are now approved for hematologic diseases, but development is also underway for solid tumors, and HER2 is included as a candidate target antigen. Safety evaluations have already been conducted for osteosarcoma, and efficacy data are expected to be obtained in the future [79].

Cancer vaccines are also being actively developed in the field of cancer immunotherapy, and clinical trials have begun for those targeting HER2 [80,81]. Most of these vaccines are being tested in breast and gastric cancers, but as their efficacy is demonstrated, it is expected that they will be tested in a wide range of HER2-positive solid tumors, as is the case with T-DXd.

## Figures and Tables

**Figure 1 ijms-24-16823-f001:**
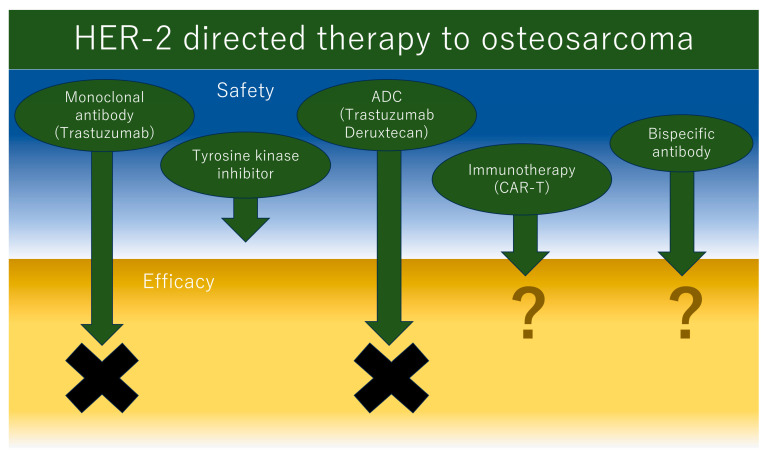
Schema of future directions of the HER2-directed therapy to osteosarcoma.

**Table 1 ijms-24-16823-t001:** Percentages of HER2 amplification, overexpression, or mutation in primary tumor types [6].

Tumor Origin	HER2 Amplification (%)	HER2 Overexpression (%)	HER2 Mutation (%)
Salivary gland	12–52	17–44	1
Lung	2–3	2.5	1–3
Breast	20	15–20	2
Stomach	11–16	20	3
Biliary tract	5–15	20	2
Pancreas	2	26	<1
Colorectum	5.8	5	2
Bladder	8.6	12.4	9
Prostate	5.8–6	10	<1
Ovary	7	27	1
Uterus	4–69	18–80	2
Cervix	0.5–14	21	3

**Table 2 ijms-24-16823-t002:** HER2-targeted drugs approved by the FDA as of 2023.

Drug	Route(s)	Indications	Year of First FDA Approval
Trastuzumab	Intravenous, subcutaneous	Breast cancer (early, advanced) Gastroesophageal cancer	1998
Lapatinib	p.o.	Breast cancer (advanced)	2007
Pertuzumab	Intravenous, subcutaneous	Breast cancer (early, advanced)	2012
Trastuzumab emtansine (T-DM1)	Intravenous	Breast cancer (early, advanced)	2013
Neratinib	p.o.	Breast cancer (early, advanced)	2017
Trastuzumab deruxtecan (T-DXd)	Intravenous	Breast cancer (advanced) NSCLC	2019
Tucatinib	p.o.	Breast cancer (advanced) Colorectal cancer	2020
Margetuximab	Intravenous	Breast cancer (advanced)	2020

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
