# Peer review of "The Future of HER2-Targeted Treatment for Osteosarcoma: Lessons from the Negative Trastuzumab Deruxtecan Results"

_ijms, 2023, doi:10.3390/ijms242316823_

Round 1
Reviewer 1 Report
Comments and Suggestions for Authors
In this article, the author reviews the development of HER2-targeted therapy in human cancers, HER2 in osteosarcoma, and clinical trial results of HER2-targted therapy in osteosarcoma, with an emphasis on the underlying reasons/mechanisms of the failure of HER2-targeted treatments for osteosarcoma, and the potential and future direction of HER2-targeted therapy in osteosarcoma. This review article is well-written, and it would add valuable information to this field. I have two very minor comments.
1. Page 1, line 30: Please change to “HER2 acts to form homodimers when binding to other HER2 molecules or heterodimers when binding to other EGFR family member”.
2. Page 3 line 101: “it is often observed that HER2-targeted therapy effective against HER2-positive breast cancer is not to other HER2-positive cancers such as T-DM1 to gastroesophageal cancer”. It is confusing. Please modify this sentence.
Comments on the Quality of English LanguageI have no problem understanding.
Author Response
Thank you for reviewing my manuscript and pointing out what needs to be corrected. I have made the following corrections.
1. Page 1, Line 30 was corrected as you noted.
2. Page 3, Line 101 was revised as follows: "there is also the case that HER2-targeted therapy effective to HER2-positive breast cancer is not effective to other HER2-positive cancers, such as T-DM1 to gastroesophageal cancer".
Reviewer 2 Report
Comments and Suggestions for Authors The review by Nakano deals about HER2 expression relevance of in osteosarcoma, both in terms of prognostic and, most importantly in this review, in terms of HER2 interest as therapeutic target. These are important issues as numerous controversies have existed in this field. Therefore, a quality review about the latest conclusions on ER2-Targeted Therapies in osteosarcoma is of importance. The author, upon an in-depth analysis and discussion of up-to-date literature, confirms that: 1) while being expressed in a number of osteosarcoma patients, HER2 expression cannot constitute a valid prognosis and 2) use of HER2-targeted treatments on patients whose osteosarcoma samples express HER2 in the form of antibodies, in particular trastuzumab, did not translate in clinical efficacy. The review is well written and logically organized. The bibliography analysis is sound and rather exhaustive. In my opinion, this review is of particular significance to the field of osteosarcoma but also beyond as HER2 overexpression has been shown to be tumorigenic and emerging data suggest efficacy of HER2-targeted therapy for HER2-amplified/overexpressing tumors across a variety of tumor types. Greater awareness of the HER2 expression impact on prognosis as well as critical evaluation of ER2-Targeted Therapies can enhance incorporation of HER2-targeted therapy into the multidisciplinary care across tumor types. Therefore, I feel this manuscript can be published as such in International Journal of Molecular Sciences.
Author Response
Thank you for reviewing my manuscript. I appreciate your comment.